# Efficiency of Sidestream Nitritation for Modern Two-Stage Activated Sludge Plants

**DOI:** 10.3390/ijerph191912871

**Published:** 2022-10-08

**Authors:** Thomas Baumgartner, Lydia Jahn, Vanessa Parravicini, Karl Svardal, Jörg Krampe

**Affiliations:** Institute for Water Quality and Resource Management, TU Wien, 1040 Vienna, Austria

**Keywords:** energy efficiency, SDE treatment, nitritation, two-stage WWTP

## Abstract

The operational costs of wastewater treatment plants (WWTPs) are mainly driven by electric power consumption, making the energy-efficient operation an all-time present target for engineers and operators. A well known approach to reduce the demand for purchased electricity is the operation of an anaerobic sludge stabilisation process. Although anaerobic digesters make it possible to recover large quantities of energy-rich methane gas, additional strategies are required to handle the increased internal return flow of nitrogen, which arises with the sludge dewatering effluent (SDE). SDE treatment increases the oxygen demand and in turn the energy required for aeration. In this study, different SDE treatment processes were compared with regard to the treatment in mainstream, sidestream nitritation, as well as nitritation combined with anammox for two-stage and single-stage WWTPs. Although SDE treatment in sidestream nitritation was found to have no effect on the energy demand of single-stage WWTPs, this concept allows the treatment capacity in the activated sludge tank to be raised, while contributing to a high nitrogen removal under carbon limitation. In contrast, SDE sidestream treatment showed great potential for saving energy at two-stage WWTPs, whereby sidestream nitritation and the further treatment in the first stage was found to be the most efficient concept, with a savings of approx. 11% of the aeration energy.

## 1. Introduction

### 1.1. Energy Assessment

Wastewater treatment includes different cleaning steps that in total require a considerable amount of energy. Wastewater treatment plant (WWTP) concepts usually encompass a mechanical treatment by screens, grit removal, and primary sedimentation to remove particular and organic matter. The subsequent biological treatment serves to remove soluble substances by microbial conversion. Especially the biological treatment is highly energy-intensive, since aeration is required to cover the oxygen demand of the bacteria involved. The energy consumption of WWTPs has been intensively analysed in recent years. Studies from different European countries reveal that the electricity consumption required for wastewater treatment can make up 1.0% of the regional energy consumption [1,2] with a specific energy demand of usually around 0.5 kWh/m^3^ wastewater [2,3]. The energy demand depends mainly on the treatment concept and the organic load, normally analysed as chemical oxygen demand (COD). In European countries, the daily amount of COD that is emitted to the sewer system ranges between 100 and 120 g COD/(PE·d). By considering a calorific value of 14 kJ/g COD [4], the energy potential of the organic material in wastewater can be calculated with 140 to 170 kWh/(PE·a). In contrast, the average energy consumption for WWTPs larger than 50,000 PE was found to be 30 kWh/(PE·a) [5], indicating that municipal wastewater contains significantly more energy in the form of organic matter than required for its treatment.

The energy demand for wastewater treatment with activated sludge is mainly driven by the sludge retention time (SRT) and the type of aeration system, whereby the energy required for the aeration can make up 60% of the total energy demand of a plant [6,7]. Moreover, the energy demand correlates with the inflow load and sludge production [8]. Figure 1 shows the proportion of the oxygen utilisation required for carbon removal (OU_C_) and the amount of carbon that is incorporated into the excess sludge (COD_ES_) in correlation to the SRT. The curves by Svardal and Kroiss [4] are based on full-scale experiences that agree with simulation results by Henze et al. [9] and can be derived from the design guideline DWA-A 131 [10]. Figure 1 illustrates that a high SRT is linked to an increased oxygen demand and a lower transfer of carbon to the excess sludge. Since the SRT for WWTPs with aerobic sludge stabilisation is prescribed with 25 d [10], the energy demand for aeration of these plants is clearly increased compared with WWTPs with anaerobic digesters. Anaerobic sludge stabilisation provides a more energy-efficient operation, since this process allows a lower SRT (meaning a lower oxygen demand) for the activated sludge tank. SRTs for WWTPs with anaerobic digesters are mostly between 12 and 14 d, whereby in that range approx. 60% of the COD is oxidised and 40% of the COD is incorporated into the excess sludge and supplied to the digester. WWTP concepts with digesters usually include primary settlers to gain a primary sludge rich in organic matter. Organic compounds of the primary and the excess sludge (hereinafter referred to as raw sludge) are degraded in the digester to energy-rich biogas. The conversion of the methane gained into energy by using combined heat and power plants (CHP) minimises the electric power purchased, making the anaerobic process a cost-effective stabilisation strategy for large-scale WWTPs.

### 1.2. SDE Treatment Concepts

The biodegradation of raw sludge leads to the release of the incorporated nitrogen and in turn to high ammonium levels in the sludge dewatering effluent (SDE). NH_4_-N levels in SDE are usually above 1000 mg/L. Since the nitrogen load in SDE can make up 20% of the incoming load, it is of high relevance to evaluate different SDE treatment concepts that can be realised on an individual WWTP.

SDE can be treated in mainstream or sidestream, i.e., directly in the activated sludge tank or in a separate tank, before it is recycled to the mainstream. Some plant concepts include a buffer tank that allows the storage and transfer of SDE into the activated sludge tank avoiding nitrogen peak loading. If there is no storage tank available, the SDE is directly returned to the activated sludge tank, where it increases the oxygen and energy demand for the aeration system significantly. The construction of dedicated treatment tanks for SDE represents an essential measure to relieve the aeration system in the biological stage and to create additional treatment capacities. A main benefit of a sidestream SDE treatment is an improved denitrification process due to a better nitrogen to carbon ratio. Since the amount of oxidised carbon is linked to the aerated tank volume, more readily biodegradable carbon is available for denitrification under lower oxygen supply. A side-effect of a reduced COD respiration under anoxic conditions is an increased adsorption of carbon to the excess sludge, which in turn raises the biogas yield during digestion. Depending on the type of SDE treatment applied in sidestream, the N to COD ratio in the influent of the mainstream tank can be also reduced in favour of denitrification.

Biological mainstream or sidestream treatment options for nitrogen rich SDE are, e.g., nitrification/denitrification [11]; nitritation/denitritation [12]; or more specific processes such as deammonification (anammox) [13,14]. The following section briefly summarises the microbial processes involved in different SDE treatment concepts.

#### 1.2.1. Nitrification and Denitrification

Nitrification is a two-step conversion process of ammonium to nitrate. The first step of nitrification is the conversion of ammonium to nitrite (nitritation) by ammonium-oxidising bacteria (AOB). Nitrite-oxidising bacteria (NOB) are responsible for the further oxidation of nitrite to nitrate (nitratation). The oxygen demand for full nitrification (nitritation and nitratation) amounts to 4.57 g O_2_/g NO_3_-N. Taking into account the biomass growth, the actual oxygen demand for nitrification amounts to 4.33 g O_2_/g NO_3_-N. Nitrification and denitrification are the conventional nitrogen removal concepts realised in the activated sludge tank (mainstream).

#### 1.2.2. Nitritation and Denitritation

SDE sidestream nitritation relies on the conversion of ammonium only to nitrite, whereby denitritation is usually performed in mainstream. Due to a lower oxygen demand for nitritation (3.25 g O_2_/g NO_2_-N), the required energy demand for aeration is approx. 25% lower than for nitrification. However, the pre-treatment by nitritation provides less oxygen in the form of nitrite for carbon removal under anoxic conditions in mainstream. Nitritation without a further conversion to nitrate can be achieved by suppressing NOBs, which are known to be more sensitive to high ammonium levels (free ammonia) and temperatures than the AOBs [15,16]. However, to avoid AOB inhibition due to free ammonia, the NH_4_-N levels should not exceed 200 mg/L [17]. The conversion of ammonium to nitrite is limited by the alkalinity; thus, only 55% of the incoming ammonium can be converted to nitrite. The pre-treated SDE is returned to the mainstream for the further nitrification of the remaining ammonium and the denitritation of nitrite by heterotrophic bacteria. In this connection, the COD demand for denitritation is approx. 40% lower compared with that for denitrification. Full-scale applications show that nitritation is characterised by a stable process control and low maintenance [18,19].

#### 1.2.3. Nitritation and Deammonification

In addition to conventional nitrogen removal, some bacteria are capable of an anaerobic ammonia oxidation. These so-called anammox bacteria require an optimal proportion of ammonium and nitrite. For that reason, the anammox process is linked to a preceding nitritation under aerobic conditions. In a second step, ammonium and nitrite are converted directly into nitrogen without the need of an organic carbon source. The following equation shows the anammox process according to Strous et al. [20].
NH_4_^+^ + 1.32NO_2_^−^ + 0.06HCO_3_^−^ + 0.13H^+^ → 1.02N_2_ + 0.26NO_3_^−^ + 0.066CH_2_O_0.5_N_0.15_ + 2.03H_2_O

The oxygen demand for the pre-treatment via nitritation and deammonification is only 1.93 g O_2_/g N and thus much lower compared with that for nitrification or nitritation. Moreover, the carbon demand is negligible, making the process attractive for WWTPs with low carbon availability or for technologies aiming for a target COD recovery such as microsieves. However, since anammox bacteria are more sensitive to environmental changes and own longer growth rates, the operation of anammox reactors requires special attention to the retention of the biomass. Operational strategies for full-scale deammonification of dewatering liquors are described by Refs. [21,22].

Table 1 summarises the oxygen demand and oxygen utilisation for carbon removal of the SDE treatment processes considered.

### 1.3. Two-Stage WWTP Design

Although most municipal WWTPs are designed as single-stage, several WWTPs are operated as two-stage activated sludge plants. The two-stage plant design relates primarily to the A/B process, which aims to separate carbon and nitrogen removal [23]. Organic carbon is partially removed in a high loaded first stage, whereby the nitrogen removal occurs mainly in the second stage. The first activated sludge tank is generally designed for carbon removal with SRTs in the range of 1 to 5 d. Due to this low SRT, most of the COD is transferred to excess sludge (55% to 60%), resulting in low oxygen consumption. The excess sludge with a high amount of biodegradable carbon ensures an increased methane yield as well as energy production, which contributes towards energy self-sufficiency. The second stage is generally designed for nitrogen removal (nitrification/denitrification) with a higher SRT. Although there is a stable operation reported for the A/B process, an unfavorable N/COD ratio could result in a lack of carbon for denitrification in the second stage. With the aim of overcoming carbon limitation in the second stage, a further development of the two-stage plant design resulted in the hybrid process. This concept relates to a sidestream that can be bypassed from the influent to the second stage in order to avoid carbon limitation and to ensure excellent nitrogen removal [24]. Generally, the operation of two-stage WWTPs is more stable, since the different organic loading rates in the activated sludge tanks can suppress the growth of filaments. A stable two-stage WWTP operation is reported for many large-scale plants (e.g., WWTP Vienna 4.0 Mio. PE; WWTP Munich I+II 3.0 Mio. PE; WWTP Frankfurt Niederrad 1.35 Mio. PE; WWTP Salzburg Siggerwiesen 680,000 PE).

The literature study indicates that there are different technologies available to treat SDE, whereby the concepts were found to affect the energy balance of a plant. Although the treatment concepts are described in earlier publications, there is so far no comparison of SDE treatment concepts available that considers municipal single- and two-stage plant design. However, from an energetic point of view, the optimal SDE treatment concept has a considerable effect on the energy demand and the operational costs. Especially large WWTPs that are often designed as two-stage plants have great potential to save energy. This paper offers valuable advice for planners and operators to decide on the most suitable SDE treatment concept. The study provides a helpful approach that can be applied for an individual plant to calculate an expected energy demand and sludge production.

## 2. Materials and Methods

### 2.1. SDE Treatment Scenarios

Different SDE treatment options were evaluated and compared by generating COD and N balances. Table 2 summarises the scenarios considered. Further SDE treatment concepts such as physical processes (e.g., ammonia stripping) or chemical processes (e.g., precipitation of magnesium-ammonium-phosphate) were not considered within this work.

Further assumptions concern the microbial conversion within the different treatment processes. The conversion of ammonium in SDE sidestream is limited by the alkalinity, whereby the effluent after nitritation usually consists of 55% nitrite and 45% ammonium. In the case of nitritation and anammox, 89% of the treated ammonium is converted to nitrogen (N_2_), 8% to nitrate and approx. 3% remain as ammonium in the pre-treated SDE. The total COD and nitrogen (TN) removal were assumed to be 95 and 80%. NH_4_-N removal was calculated with 100%.

### 2.2. COD and N Balances

Mass balances are well-known approaches to illustrate differences in the mass flow of a system. Figure 2 shows the COD and N mass balances for the single-stage WWTP concept. The following calculations are based on specific data for municipal WWTPs. The first assumption concerns the inflow loads, which are defined as 120 g COD/(PE·d) and 8 g N/(PE·d). This specific nitrogen load is common for WWTPs > 10,000 PE [25]. All mass balances consider a total nitrogen removal rate of 80%, which corresponds to an effluent load of 1.6 g N/(PE d). COD removal was calculated as 95%. COD and N loads that do not end up in the effluent must be removed by oxidation processes, biomass growth, or methane production.

The mechanical treatment step by primary sedimentation considers a COD and N removal of 30% and 10%, which end up in the primary sludge treated afterwards in the digester. Since the removal rate during primary sedimentation is independent of the SDE treatment concept, the COD and N loads to the digester and activated sludge tank (AST) are the same for all scenarios considered. The same applies to the excess sludge production during biological treatment, since all scenarios are based on a COD inflow load of 84 g COD/(PE·d) and a SRT of 15 d. During biomass growth, approx. 7% of the removed COD is incorporated as nitrogen into the biomass [10].

The primary and excess sludge are treated in anaerobic digesters, whereby the output after digestion was assumed to be 30 g COD/(PE·d) and 1.8 g N/(PE·d), according to the experience at real plants (especially in Austria and Germany) and to modelling results [26]. The methane produced during digestion is converted by CHP with an electrical efficiency in the range of 30% to 40% (3 kWh_el_/m^3^ CH_4_). For this study, the CHP efficiency was assumed to be 30%.

Incorporated nitrogen is released during digestion. Since the anaerobic degradation is the same, the nitrogen load in SDE is similar. Differences appear for the nitrogen load returned to the activated sludge tank, which depend on the applied SDE treatment concept. Thus, the energy demand for nitrification is driven by the nitrogen load after SDE treatment.

Figure 3 shows the COD and N mass balances for the two-stage WWTP concept. The same inflow loads were considered as for the single-stage design. However, there are differences with regard to the first-stage and the excess sludge production. COD removal in the first stage was calculated with 60%. Thus, the COD load to the second stage was 33.6 g COD/(PE·d) for all scenarios considered. In this context, the excess sludge produced in the activated sludge tank as well as the incorporated nitrogen resulted in the same amount. SRT in the first stage was defined to be 5 d. Since the biomass yield depends on the sludge retention time, the proportion of COD transferred into excess sludge accounts for 60% and 40% for oxidation (Figure 1). The amount of already oxidised nitrite or nitrate returned from SDE treatment to the first-stage reduces the COD load that has to be oxidised and in turn the aeration energy required. Thus, there appeared differences in the energy demand for aeration in the first and second stage.

### 2.3. Further Assumptions

The energy demand for the biological stage is linked to the efficiency of the aeration system. For tapwater, the standard aeration efficiency (SAE) of fine-bubble aeration systems is usually between 3.0 and 4.0 kg O_2_/kWh [27]. The SAE in wastewater is well below this value and depends on different factors, e.g., aeration system, dissolved salts, detergents, reactor configuration, dry solid content, and SRT [28]. The α-factor is the ratio of the oxygen transfer into tapwater to the oxygen transfer into wastewater and a relevant design parameter for aeration systems. The α-factor should be considered carefully for the different treatment tanks, since it determines the energy demand for aeration. The main parameter affecting the α-factor seems to be the SRT, which is directly associated with the biomass concentration in the activated sludge tank [29]. Frey [27] reported a low SRT in the first stage of two-stage activated sludge plants with α-factors in the range of 0.3 to 0.4. For nitrifying and denitrifying plants (SRT from 12 to 14 d), the α-factor ranges from 0.5 to 0.7. Postulating an α-factor of 0.7 for separate SDE treatments, as indicated in practice, provides an additional advantage by reducing electricity demand for aeration aiming nitrification/nitritation. Nitrite produced in sidestream with a more favourable α-factor (lower electricity consumption) is used to remove a part of the COD load in mainstream through denitritation, avoiding that this COD is oxidisied aerobically by O_2_ provided with less efficiency (lower α-factor). The energy saving is much higher at two-stage plants due to the lower α-factor in the first-stage. Table 3 summarises α-factors that were considered for the different treatment tanks and plant configurations.

## 3. Results

Table 4 summarises the results of the SDE treatment scenarios. The corresponding mass balances are shown in the Appendix A. The nitrogen load in the SDE of the single-stage WWTP was calculated with 1.2 g N/(PE·d), which corresponds to 15% of the incoming nitrogen load. The calculations presented are based on a TN removal of 80%. After SDE sidestream treatment, the NH_4_-N load to the activated sludge tank clearly decreased, resulting in a lower energy demand for aeration. However, the total energy demand for aeration in the separate SDE tank and the activated sludge tank was in total approx. 12.2 kWh/(PE·a), nearly the same for all scenarios considered. Although some publications claim a lower oxygen and thus a lower electricity demand for the anammox process [30,31], the results of our study revealed that the lower electricity demand is not the case when considering the total aeration demand. The results illustrated that the overall energy consumption for the biological treatment at single-stage WWTPs can not be affected by the implementation of a SDE sidestream treatment. Similar results were found for the energy recovery from biogas utilisation, which was equal for all scenarios with approx. 14.3 kWh/(PE·a). However, a significant difference was found for the organic carbon demand for denitrification. The highest carbon demand accounted for scenario 1 (mainstream) with 4.8 kg COD/(PE·a), whereby the lowest carbon demand of 3.7 kg COD/(PE·a) was found for SDE nitritation and anammox. SDE pre-treatment with nitritation resulted in a 6% lower carbon demand. In the case of a limited COD availability or by focusing a target COD removal, SDE sidestream treatment offers suitable handling to ensure extensive nitrogen removal and to relieve the aeration system in the activated sludge tank. Besides the lower carbon demand, the results of the mass balances indicated that the energetic advantage of sidestream SDE treatment of single-stage WWTPs is less. Moreover, the operation of a sidestream treatment is linked to additional construction costs and the need of technical equipment (e.g., aeration system with blowers and stirrers). Control and handling of SDE sidestream treatments can be challenging when there arise foaming problems that can only be controlled by dosing anti-foaming agents. Operators and designers should carefully check the site-specific benefits for sidestream SDE treatments on single-stage WWTP design.

In the second part of the study, different SDE treatment concepts were considered for the two-stage plant design. The nitrogen load in the SDE of the two-stage WWTPs was calculated with 1.7 g N/(PE·d), which corresponds to 21% of the incoming nitrogen load. Sidestream concepts (scenario 5 and 6) consider the return of the pre-treated SDE to the first stage of a two-stage WWTP, which significantly reduces the energy demand for aeration in the first stage due to denitritation. Additionally, the higher oxygen transfer rate in the sidestream tank has a positive effect on the total energy balance compared with the single-stage plant design. Overall, the energy demand for aeration of the two-stage plant is considerably lower compared with the single-stage WWTP operation (−9.3%). The lowest energy demand for aeration was found with 9.8 kWh/(PE·a) for the scenario with sidestream nitritation of SDE. Moreover, energy savings are possible with SDE sidestream nitritation and anammox. Compared with SDE treatment in mainstream, the SDE treatment with nitritation allows energy savings of approx. 8%; 3% of energy savings are possible for nitritation and anammox. The results of the two-stage plant design indicate that differences in the aeration energy of the SDE treatment concepts considered can make up 0.3 to 0.9 kWh/(PE·a). Since the specific energy consumption is usually in the range of 20 to 30 kWh/(PE·a), the total energy savings can account for 4.5% or more than 90 MWh/a for WWTPs larger 100,000 PE.

A side-effect of the anoxic respiration and the low SRT in the first stage is an increased transfer of COD to the excess sludge with the benefit of higher biogas production in the digester. Compared with the single-stage WWTP operation (scenarios 1 to 3), the anaerobic stabilisation at two-stage WWTPs (scenarios 4 to 6) enables an overall higher gain of energy from anaerobic sludge stabilisation (+33%). The highest possible energy recovery from biogas was calculated with 19.9 kWh/(PE·a) for SDE sidestream nitritation, followed by nitritation and anammox with 19.4 kWh/(PE·a). The results indicate that sidestream SDE treatment allows an additional gain of energy from biogas up to 90 MWh/a considering a two-stage WWTP with a design capacity of 100,000 PE. Although SDE treatment in sidestream is attractive for saving energy and increasing the biogas yield, it was found in earlier studies that there can appear considerable N_2_O emissions due to higher nitrite levels and loads. Especially sidestream nitritation was found to increase N_2_O emissions of a WWTP [32]. Since N_2_O is a climate relevant greenhouse gas, increased emissions affecting the CO_2_ footprint of a WWTP and should also be considered.

Moreover, the calculations represented provide information about the carbon demand of the SDE treatment concepts. The lowest carbon demand for denitrification was found for the SDE treatment via nitriation and anammox with 3.16 kg COD/(PE·a), which is approx. 30% to 35% lower compared with the alternative scenarios. However, the carbon efficiency of the deammonification process presents no benefit, as the first stage needs to be operated under aerobic or anoxic conditions. Even though it is stated in many references that the deammonification of SDE reduces the energy demand of municipal WWTPs considerably, this is not the case for typical municipal wastewater where the oxygen demand for the carbon removal exceeds the oxygen demand for the nitrification. Moreover, there is an increased process control required in order to provide enough COD for denitrification in the second-stage. High N/COD ratios can restrict denitrification in the second-stage due to a lack of carbon and must be considered carefully.

The results confirmed that the concept of two-stage WWTPs is advantageous for an energy-efficient operation with an increased energy self-sufficiency. Nitritation in SDE pre-treatment tanks and the use of nitrite to substitute oxygen in the first stage of a two-stage WWTP appears to be the most advantageous SDE treatment concept for saving aeration energy. Nevertheless, it should be mentioned that different process settings and operational conditions can enable another SDE treatment option to the more favorable concept for an individual plant. So far, publications reporting full-scale operation that allow verifying the results presented are scarce. Full-scale experiences from the implementation of a SDE sidestream nitritation on the second-stage WWTP Kirchbichl were reported by Ref. [33]. The authors observed a stable nitritation after the startup of the process and temporary problems with foam. However, due to changes in the digester operation and internal return flows, it was not completely possible to evaluate the energetic effects out of the received data for the restricted experimental period. For the background of changeable operational conditions, mass balances have been proven to be an efficient tool to forecast energetic effects. The presented data serve to explain how mass balances can be applied with regard to assess SDE treatment.

## 4. Discussion

The goal of the study was to provide a comparison of different SDE treatment concepts by considering single- and two-stage plant design. The calculations performed by mass balances are based on specific data received from benchmark studies and full-scale experiences. Operators and engineers can use the described method to evaluate the best suitable SDE treatment concept for an individual WWTP by using plant-specific loads and data. The results presented demonstrated that the SDE sidestream treatment at single-stage WWTPs does not display an energetic advantage for one of the investigated treatment methods, since the energy demand for aeration was approx. 12.2 kWh/(PE a) for all scenarios considered. Moreover, the energy potential of the biogas was the same with 14.3 kWh/(PE·a). However, the deammonification process had a higher carbon efficiency with a 23% lower carbon demand compared with mainstream SDE treatment, which is a great benefit for single-stage activated sludge plants with high effluent requirements regarding nitrogen removal or a limited capacity of the aeration system. Overall, it was found that the two-stage WWTP operation allows an approx. 30% higher biogas production compared with single-stage WWTPs, since the anoxic respiration of heterotrophic bacteria and low SRT lead to a higher amount of COD transferred to sludge. This in turn reduces the energy demand for aeration considerably, whereby the energy demand for aeration is approx. 13% lower by considering the mainstream SDE treatment. SDE pre-treatment in sidestream via nitritation and the further treatment of nitrite-rich effluent in the first stage of a two-stage WWTP reduces the oxygen demand for COD respiration and is therefore the most favourable strategy to achieve an energy-efficient wastewater treatment with savings of approx. 8% of aeration energy.

## Figures and Tables

**Figure 1 ijerph-19-12871-f001:**
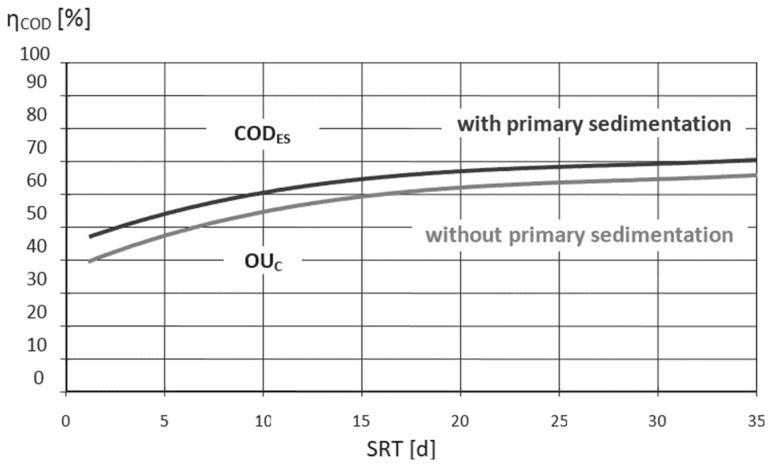
COD in excess sludge (COD_ES_) and the oxygen utilisation for carbon removal (OU_c_) in correlation to the SRT [4].

**Figure 2 ijerph-19-12871-f002:**
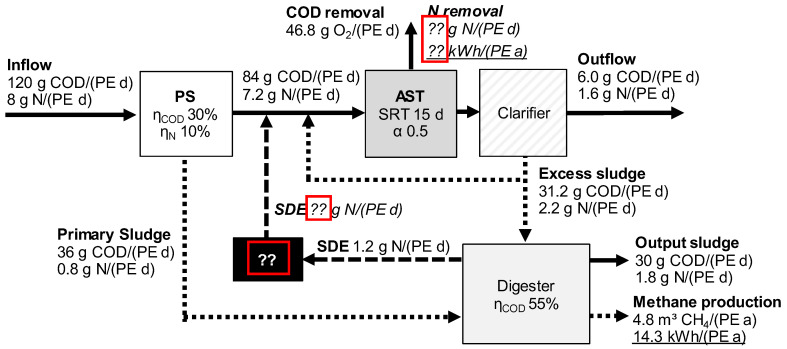
COD and N balances for single-stage activated sludge tanks.

**Figure 3 ijerph-19-12871-f003:**
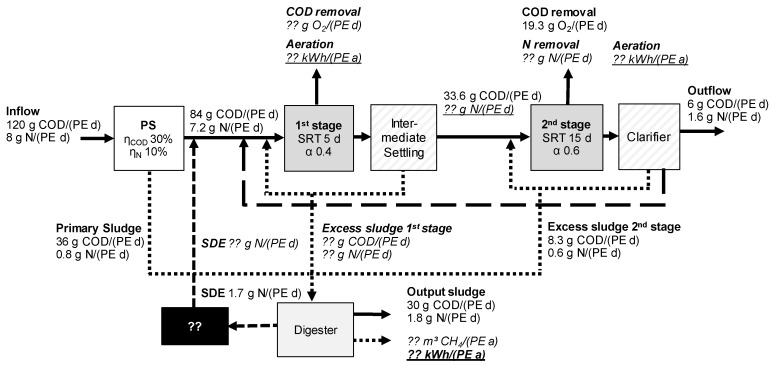
COD and N balances for two-stage activated sludge tanks.

**Table 1 ijerph-19-12871-t001:** Oxygen demand and oxygen utilisation for carbon removal for denitrification (OU_DN_).

SDE Treatment Concepts	Oxygen Demand (g O_2_/g N)	OU_DN_ for TN Removal (g O_2_/g N)	Total (g O_2_/g N)
nitrification/denitrification	4.33	2.86	~1.5
nitritation/denitritation	3.74	2.23	~1.5
nitritation/anammox	1.93	0.40	~1.5

**Table 2 ijerph-19-12871-t002:** Scenarios for the SDE treatment in single- and two-stage WWTPs.

Scenario	Plant Concept	Sidestream	Mainstream
1	single-stage	-	nitrification/denitrification
2	single-stage	nitritation	denitritation
3	single-stage	nitritation/anammox	-
4	two-stage	-	nitrification/denitrification
5	two-stage	nitritation	denitritation in 1st stage
6	two-stage	nitritation/anammox	-

**Table 3 ijerph-19-12871-t003:** α-values considered for different treatment tanks.

Configuration	Treatment Tanks	α-Factors
Single-stage	activated sludge tank	0.5
Two-stage	first-stage	0.4
Two-stage	second-stage	0.6
SDE treatment	sidestream tank	0.7

**Table 4 ijerph-19-12871-t004:** Energy for aeration, energy from biogas utilisation, and OU_DN_ for TN removal of 80% in single-stage and two-stage WWTPs with different SDE treatment options.

Scenario		Energy from Biogas (kWh_el_/(PE·a))	Energy for Aeration (kWh_el_/(PE·a))	OU_DN_ (kg COD/(PE·a))
single-stage	1	14.3	12.3	4.80
	2	14.3	12.2	4.53
	3	14.3	12.2	3.70
two-stage	4	19.0	10.7	4.80
	5	19.9	9.8	4.41
	6	19.4	10.4	3.16

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
