# Peer review of "Efficiency of Sidestream Nitritation for Modern Two-Stage Activated Sludge Plants"

_ijerph, 2022, doi:10.3390/ijerph191912871_

Round 1
Reviewer 1 Report
Some suggestions I think the authors should consider and list below:
1. the methodology in this version is too simplified and I suggest the authors describe it in the revised version.
2. I suggest the study includes a robust check to reconsider the consistency of the outcomes.
3. the conclusion part is too simple now, suggest the author should focus on the study motivation or the background. Now the description in the original manuscript is more like to hastily explaining the results of the study.
4. some grammar errors in the sentence should to recheck in the manuscript.
Author Response
Dear Reviewer,
thank you for reviewing this paper and for providing helpful comments and suggestions. We considered the comments in the following way:
- Methodology is now described more in detail in the revised manuscript. We hope the method part is now more clear and comprehensible (Line 165-200).
- Outcomes were checked twice.
- Novelty point and purpose of the study were added at the end of the introduction part (Line 140-148).
- We also highlight the motivation in the conclusion of the study in order to provide a comprehensible comparison of SDE treatment concept that can be applied for individual plant.
- The grammar of the manuscript was checked again by a native speaker.
We hope that the rework improves the quality of the paper.
Kind regards
Reviewer 2 Report
The authors had investigated the different SDE (Sludge Dewatered Effluent) treatment through three different methods, including the mainstream, the nitritation, and the anammox. There are some collected results from this investigation to determine the energy consumption in the field of wastewater treatment method. However, there are some weakness points in this research that the authors should address them to improve their research paper.
- The concept “WWTP” must be explained firstly in the abstract.
- The novelty points in the wastewater treatment method must be indicated and analyzed with other methods to determine the energy consumption.
- The main part of this paper completely only stated the previous research results. The state-of-the-art methodologies have not been seen from this research.
- The determination of the energy consumption of the wastewater treatment is very necessary to verify through the experimental works. However, the authors have not still conducted this one.
- The discussion is lacked in this research.
Author Response
Dear Reviewer,
thank you for reviewing this paper and for providing helpful comments and suggestions. We considered the comments in the following way:
- A short description of the WWTP concept has been added in the introduction part (Line 23‑28). We decided not to add it in the abstract, since the abstracts should only focus on the most relevant information of the presented work.
- Novelty point and purpose of the study were added at the end of the introduction part (Line 140-148).
- Methodology is now described more in detail in the revised manuscript. We hope the method part is now more clear and comprehensible (Line 165-200).
- We also highlight the motivation in the conclusion of the study in order to provide a comprehensible comparison of SDE treatment concept that can be applied for individual plant.
- Verification with full-scale is very challenging, since the operation without and with SDE has to be under comparable process conditions. A discussion part was extended by also giving advice with regard to N2O emissions.
- All the cited references were checked for their relevance. The manuscript now includes the most relevant references. Some few reference were added since the discussion part was extended.
- The grammar of the manuscript was checked again by a native speaker.
We hope that the rework improves the quality of the paper.
Kind regards
Round 2
Reviewer 2 Report
The authors had addressed some comments from this reviewer. However, there are some drawbacks that the authors should address them to improve their research paper.
1. The authors must provide the clear manuscript with the highlighted changes for reading easily than using the track changes tool of MS Word.
2. The contribution of this research as well as the novelty points must be indicated in the introduction part.
3. The conclusion must summary the collected results and their significance for treating the wastewater technology.
Author Response
Dear Reviewer,
we completely reworked the Result and Conclusion part and provide a MS word document with track change. Introduction part includes now novelty points and the target of the work.
We hope that the changes made will help to improve the quality of the paper.
Thanks again for your time and support by providing helpful suggestions and comments.
Kind regards